# Natural Rubber Composites Using Hydrothermally Carbonized Hardwood Waste Biomass as a Partial Reinforcing Filler—Part II: Mechanical, Thermal and Ageing (Chemical) Properties

**DOI:** 10.3390/polym15102397

**Published:** 2023-05-21

**Authors:** Jelena Lubura, Olga Kočková, Beata Strachota, Oskar Bera, Ewa Pavlova, Jelena Pavličević, Bojana Ikonić, Predrag Kojić, Adam Strachota

**Affiliations:** 1Faculty of Technology Novi Sad, University of Novi Sad, Bulevar cara Lazara 1, 21000 Novi Sad, Serbia; 2Institute of Macromolecular Chemistry, Czech Academy of Sciences, Heyrovskeho nam. 2, CZ-162 00 Praha, Czech Republic

**Keywords:** natural rubber composites, bio-sourced raw materials, hydrochar, hydrothermal carbonization, carbon black, mechanical properties, degradation stability

## Abstract

Natural rubber composites were reinforced by the co-fillers ‘hydrochar’ (HC), obtained by hydrothermal carbonization of hardwood sawdust and commercial carbon black (CB). The content of the combined fillers was kept constant while their ratio was varied. The aim was to test the suitability of HC as a partial filler in natural rubber. Due to its larger particle size and hence smaller specific surface area, large amounts of HC reduced the crosslinking density in the composites. On the other hand, due to its unsaturated organic character, HC was found to display interesting chemical effects: if it was used as the exclusive filler component, it displayed a very strong anti-oxidizing effect, which greatly stabilized the rubber composite against oxidative crosslinking (and hence embrittlement). HC also affected the vulcanization kinetics in different ways, depending on the HC/CB ratio. Composites with HC/CB ratios 20/30 and 10/40 displayed interesting chemical stabilization in combination with fairly good mechanical properties. The performed analyses included vulcanization kinetics, tensile properties, determination of density of permanent and reversible crosslinking in dry and swollen states, chemical stability tests including TGA, thermo-oxidative aging tests in air at 180 °C, simulated weathering in real use conditions (‘Florida test’), and thermo-mechanical analyses of degraded samples. Generally, the results indicate that HC could be a promising filler material due to its specific reactivity.

## 1. Introduction

**Natural rubber** products are commonly used in a great variety of applications, and generally, they are reinforced by stiff nanofillers, typically by **carbon black** (CB) [1,2,3,4]. The production of the latter involves thermal oxidative processes, which typically entail partial combustion of liquid aromatic hydrocarbon oil, in the oil furnace process, within the temperature range from 1400 to 2000 °C [5]. The filler’s key defining properties include its size, structure, surface activity, specific surface area, and its particle shape. The filler improves the static and dynamic behavior of the rubbers. It affects in different ways the rubber composites’ strength, elongation and break, toughness, stiffness, processability, as well as energy dissipation and friction [6]. The properties of natural rubber products also depend on the filler distribution within the matrix, as well as on filler characteristics, such as chemical structure, surface activity, particle size, and shape [7]. Functional groups on the surface of filler particles can strongly alter the filler–matrix interactions, and thus change the strength of the reinforcing effect [8].

In view of the recent development of the legal and economic situation, the rubber industry currently seeks **new, naturally growing raw materials**. This also concerns fillers for natural rubber. **Exploiting production waste**, such as hardwood sawdust studied in this work, in the role of a precursor of filler for natural rubber, would be of significant economic advantage, independently of a specific situation. On the other hand, the mechanical and thermo-mechanical properties of rubber which is filled with **new bio-fillers**, have to fulfill high market requirements, similar to the traditional CB filler.

In this context, so-called **bio-carbon** filler materials enjoy considerable research interest. According to Bahl [9], using lignin-based materials as hybrid fillers along with carbon black can effectively reduce the viscoelastic loss in rubber compounds. Incorporating these materials can increase the resistance of the so-filled rubbers against aging and flex-cracking, and it also can improve the composites’ processability, as reported by Yu [10]. However, Behazin [11] found that bio-carbons with a high ash content do not improve the effect of the (combined) filler phase. Nonetheless, bio-carbon with low ash content (no more than 2%) was found to be suitable to partially substitute carbon black as filler for styrene–butadiene rubber. Peterson [12] reports that substituting 25% or 50% of carbon black with bio-carbon can improve tensile strength, elongation at break, and toughness compared to similar composites filled exclusively with carbon black.

A promising special method for obtaining bio-sourced carbon fillers is **hydrothermal carbonization** (HTC), to which considerable efforts were dedicated [13] and which yields valuable carbonaceous products called “hydrochar” (HC) [14,15,16,17]. In the course of HTC, the biomass feed is heated in an oxygen-free environment in the presence of subcritical water, and its oxygen and hydrogen content is lowered by dehydration and decarboxylation reactions [18,19]. The main advantages of HTC are its cost-effectiveness and lower energy consumption compared to classical pyrolysis routes used to produce biochar [20]. Importantly, no drying of the biomass feed is needed. Moreover, a great variety of feedstocks can be used in the HTC process, such as cellulose [21], corn stack [22], tomato-peel waste [23], sewage sludge [20], and even (sorted) municipal solid waste [24]. The precise physicochemical properties of the obtained hydrochar depend on the feedstock type as well as on HTC process parameters, such as carbonization time, process pressure, and residence time [25]. An advantage is that the easily applicable HTC is considered to be a non-toxic and environmentally friendly process [26].

In a previous paper, which is the first part of the wider presented work [27], the authors applied their own brief HTC treatment method to sawdust originating from the oak tree. A comprehensive characterization of the so-obtained HC filler was undertaken, including some aspects of blending the obtained HC with natural rubber.

**The aim of the authors’ present work** was to **assess the suitability of the bio-sourced hydrochar (HC)**, obtained according to [27], as a filler (or co-filler) in natural rubber and similar elastomers. For this purpose, a standard rubber recipe was modified by partially (and even fully) substituting the classical CB filler with HC while the loading of the co-fillers phase was kept constant.

Basic **mechanical and thermo-mechanical properties** were analyzed (tensile, hardness tests, DMTA, swelling, and mechanical tests in swollen state) in order to assess the effect of HC on **permanent and reversible crosslinking**.

**The chemical effects of HC** were studied by several methods:-Long-time vulcanization experiments were carried out in order to compare the behavior of the rubber composites filled with HC/CB with one of the filler-free rubber recipes, as well as to assess the kinetics of crosslinking and degradation in more detail.-TGA was employed to compare the tendency of the composites and of the neat matrix to thermolysis (in nitrogen) and to oxidative degradation (in air);-Thermo-oxidative aging test at a constant, elevated temperature in air (180 °C) was used to relatively quickly characterize differences in resistance to oxidation (to degradation, or to embrittlement);-Weathering in real-use “hot and humid” conditions also was simulated (industrial ‘Florida test’).

The results of all the above methods indicated that HC might be an interesting filler in organic polymers due to its distinct anti-oxidizing effect combined with a reasonably high reinforcing ability.

## 2. Materials and Methods

### 2.1. Materials

Natural rubber Standard Vietnamese Rubber CV60 (Vietnam Rubber Group), as well as the following additives: *N*-isopropyl-*N*′-phenyl-p-phenylenediamine (IPPD), stearic acid, zinc oxide (ZnO), sulfur, and *N*-cyclohexylbenzothiazol-2-sulfenamide (CBS), were all kindly obtained from Edos (Zrenjanin, Republic of Serbia), and used as received without further purification (for purity see further below Section 2.3. Composition of the rubber mixtures). Carbon black N330, the conventional filler component with a typical particle size of 30–50 nm, was purchased from Nhumo (Altamira, Mexico). The preparation of hydrochar, used as a potential replacement for CB, is described in detail in the next section.

### 2.2. Preparation of the Hydrochar Filler

In order to obtain the hydrochar filler, hydrothermal carbonization treatment of hardwood waste biomass (sawdust originated from the oak tree, cellulose content: 91 wt.%, lignin content: 9 wt.%) was performed at 300 °C, under autogenous pressure of 86.6 bar, with the process duration of 30 min. The obtained hydrochar consisted of relatively large particles and their agglomerates. In the next step, the raw product was ground in a planetary mill (model Mono Mill Pulverisette 6), from Fritch (Idar-Oberstein, Germany), at 200 rpm for 5 min. The grounded particles were then sieved from 500 to 800 μm, and thereafter, they were washed with hot deionized water until the dark leachate (containing degraded lignin and cellulose fragments) was fully extracted. Finally, the product was dried in an oven for 24 h. The total hardwood weight loss after HTC treatment and sieving was 62.3%, which means that the hydrochar **yield was 43.8%**.

### 2.3. Composition of the Rubber Mixtures

The recipe for obtaining the studied natural rubber composite samples is presented in Table 1, namely the phr amounts of all constituents and additives.

As presented in Table 1, the additives used in this work are *N*-isopropyl-*N*′-phenyl-p-phenylenediamine (IPPD; anti-oxidant and anti-ozonant), degree of purity 97%; zinc oxide (ZnO; simple vulcanization accelerator), degree of purity 99.9%; stearic acid (dispersing agent for ZnO), degree of purity 97%; sulfur (crosslinker), degree of purity 99.999%; and *N*-cyclohexylbenzothiazol-2-sulfenamide (CBS; vulcanization accelerator with delayed action), degree of purity 95%. A necessary component in any technical rubber is the reinforcing filler. In this work, two types of rubber filler were used: hydrochar (HC) and carbon black (CB). The total filler amount was kept at 50 phr (according to ASTM-D1765), while their ratio was varied (10, 20, 30, 40, and 50 phr of CB plus 40, 30, 20, and 10 phr of HC, respectively). The content of the remaining additives was kept constant.

The prepared rubber samples were labeled according to the carbon black content (classical filler in the filler mixture), as shown in Table 2. The matrix presented in Table 2 was formulated in the absence of fillers—co-filler fractions in wt. % and in % *v*/*v* also are listed in Table 2.

### 2.4. Mixing and Vulcanization Procedure

The mixing procedure included three steps: (1) mixer preparation: idle run, conditioning at 90 °C, (2) component preparation: mixing of neat natural rubber at several speeds until constant process parameters; and (3) component mixing: admixing of the remaining components from Table 1. The mixing procedure was performed using the Laboratory mixer Haake Rheomix (model 600, Thermo Fisher Scientific, Waltham, MA, USA), modified with a drive unit Haake Rheocord EU-5. The employed three-step mixing procedure was identical to the one described in detail in previously published work [28].


**Vulcanization:**


The samples of the rubber composites were vulcanized following the ISO 37 standard by pressing the rubber sheets for 15 min at 150 °C at atmosphere pressure. Subsequently, the vulcanized sheets (thickness: 3 mm) were left to relax for 24 h at room temperature. Thereafter, the samples were cut into specimens of a dumbbell shape.

### 2.5. Rheology of Vulcanization

The rotorless rheometer MDR-A, supplied by Beijing Rade Instrument Co., Ltd., Beijing, China, was used for monitoring the vulcanization process. For each prepared sample, the rheological tests were performed for 2 h at 150 °C.

### 2.6. Mechanical Properties

Hardness, tensile strength, and elongation at break were measured to determine the prepared samples’ basic mechanical properties.

**Hardness** was measured with a Shore A durometer, model Zwick 3100, manufactured by Zwick (now Zwick Roell, Ulm, Germany), in accordance with the ISO 7619-1 standard. All the measurements were repeated three times, and the mean hardness value was obtained and expressed in Shore A units.

**Tensile tests:** The samples’ tensile strength and elongation at break were measured on a dynamic Rade extensometer, RT5K-2, manufactured by Beijing Rade Instrument Co., Ltd., Beijing, China, according to ISO 37 (crosshead speed: 500 mm/min). The tests were repeated five times for each vulcanized sample, and the average value was taken result.

### 2.7. Thermo-Mechanical Properties (DMTA)

Dynamic-mechanical thermal analysis (DMTA) was performed on an ARES G2 apparatus from TA Instruments (New Castle, DE, USA, now part of Waters, Milford, MA, USA), in oscillatory mode, at a deformation frequency of 1 Hz. Deformation amplitude ranged from 0.01 to 3.5% and was adjusted automatically using the AutoStrain function. The investigated temperature range was from −100 to 180 °C, while the heating rate was 3 °C/min. A constant axial tension force of 5 g (49.1 mN) was upheld in order to prevent sample bending due to thermal expansion.

The samples had standardized geometry: rectangular platelets of the dimensions: 30 mm × 6 mm × 3 mm.

Swollen samples, which were also characterized by DMTA, had different geometries, ranging between 44 mm × 8.8 mm × 4.4 mm and 49.5 mm × 9.9 mm × 5.0 mm, depending on the final swelling degree. In contrast to dry samples, the swollen ones were measured only at *T* = const. = room temperature (=25 °C). The swollen samples were taken out of the swelling bath immediately prior to the DMTA test.

Output: Storage modulus, loss modulus, and the loss factor (tan *δ*) were measured as a function of temperature (in the case of swollen samples: at constant room temperature as a function of time). The glass transition temperature was defined as the temperature of the maximum value of the loss factor (tan *δ*).

### 2.8. Swelling Tests

Swelling of the natural rubber nanocomposite samples in toluene at room temperature was followed for 24 h (1440 min). For this purpose, samples weighing ca. 540 mg (dimensions: 30 mm × 6 mm × 3 mm) were immersed into a 40 mL toluene bath, and the weight of the swelling samples was measured after 5, 10, 15, 25, 35, 55, 75, 105, 135, 195, 255, 315, 375, and 1440 min. The swelling degree was calculated by applying the following Equation (1):(1)$$Swelling\;degree,\;\%=\frac{w_{s}-w_{p}}{w_{p}}\times 100\%$$
where *w_p_* is the sample’s initial mass (grams), and the *w_s_* is the mass of the swollen sample (grams) after a given swelling time [29].

After 1440 min (24 h) of swelling, the so-treated rubber samples were forwarded to dynamic-mechanical analysis at room temperature to investigate the influence of swelling on the storage and loss moduli of the samples.

### 2.9. Thermogravimetric Analysis (TGA)

TGA analyses were performed using a Pyris 1 TGA thermogravimetric analyzer (from PerkinElmer, formerly Perkin-Elmer, Waltham, MA, USA), in a temperature range from 35 to 800 °C, at the heating rate of 10 °C/min under a constant gas flow of 20.0 mL/min. All the samples (weight: 5–10 mg) were analyzed in nitrogen as well as in air. As output, the mass = *f*(temperature), and its derivative curve was registered.

### 2.10. Rubber Aging

**Thermo-oxidative aging test in air at 180 °C**:

Small pieces were cut from the studied vulcanized samples, namely 4 pieces per material composition. Approximate specimen size was 30 mm × 6 mm × 3 mm, corresponding to a weight of ca. 0.54 g. The so-prepared samples for the thermo-oxidative aging test were hung on a copper wire in an oven with forced air circulation, at *T* = 180 °C, for aging times of 30, 60, 180, and 360 min. The procedure was similar to literature works [30] and was directly based on the authors’ previous experience with achieving relatively rapid thermo-oxidative aging [31]. After each aging time, one specimen of every studied material was removed from the oven, and the samples’ mass was measured to determine the mass loss. Next, each removed specimen was analyzed by DMTA in order to evaluate changes in its mechanical and thermo-mechanical properties after the given aging period.

**‘Florida test’ of tires’ aging**:

The aging of the studied rubber nanocomposites in a moist and hot climate, which is caused by the weather and by daylight UV/vis-irradiation, was assessed by means of realistic artificial weathering so that the environmental conditions were simulated in a shorter experimental time. The weathering (“accelerated aging”) was performed according to the “PV 3930” method (‘Florida test’) developed by Volkswagen (Wolfsburg, Germany), which relies on the guideline ISO 4892-2.

Experiment parameters for the ‘Florida test’ were as follows:

The test was based on two alternating periods, dry and wet:-a dry period proceeded for 102 min under UV irradiation: wavelength = 340 nm (using emission filter), the intensity of irradiation: 0.50 W/m^2^;-a wet period (sprinkling in the dark) proceeded for 18 min.

Both periods combined (120 min = 2 h in total) corresponded to one simulated day.

Specimen chamber temperature was kept at 35 °C, while the relative humidity was maintained at 60%. The black standard temperature (BST) measured at these conditions was 65 °C.

Sets of samples were subjected to total test duration times of 7 days (corresponding to 2.75 real months) and 25 days (corresponding to 10 real months).

## 3. Results and Discussion

### 3.1. Rubber Recipe Mixing, Morphology, and Vulcanization

This work aimed to investigate the potential of bio-sourced hydrochar (HC) for the role of carbonaceous filler in elastomeric composites, specifically in composites with natural rubber as a matrix, similar to rubber composites used in tires. The hydrochar (HC) was obtained from sawdust waste via the modern hydrothermal carbonization method (HTC), adjusted by the authors, and followed by ball milling, as described in detail in the first part of this broader work [27]. For evaluating the HC-based nanocomposites, a classical tire rubber recipe (natural rubber + filler phase + vulcanization accelerator ZnO + stearic acid as a dispersion agent for ZnO + CBS as vulcanization accelerator with delayed action + sulfur as crosslinker + IPPD as anti-oxidant and anti-ozonant; details: see Table 1 in the Experimental Part, Section 2.3), which contained 50 phr (31.5 wt.%) of carbon black (CB) as the filler phase, was modified. CB was partly or completely substituted by HC while keeping the content of the combined fillers constant at 50 phr (see Figure 1). The co-fillers’ ratio HC/CB varied between 0/50 and 50/0, in steps by 10 phr. The filler-free vulcanized natural rubber matrix, containing all additives except the fillers, was prepared and characterized as a reference material. The codes of the sample names, the weight, as well as the volume fractions of the fillers are listed in Table 2 in the Experimental Part (Section 2.3).

In Figure 1, the **morphology** of the prepared rubber composites (and of the filler-free ‘neat’ matrix) is compared (all the samples are in the vulcanized state), as observed by transmission electron microscopy (TEM). It can be noted that the co-fillers strongly differ in their size: HC (irregular particles: relatively light grey in Figure 1c,d, and darker in Figure 1b) displays sizes between 0.5 and 3 µm, while the diameter of CB (very dark small spheres) ranges from 30 to 60 nm. Both **nanofillers were extensively characterized** in the mentioned previous work [27]. The BET surface areas were found to be 77.8 m^2^/g and 21.4 m^2^/g for CB and HC, respectively. These results suggested a significant porosity of the much larger HC filler particles since the estimated (in the supplementary information file of [27]) external surface area values (without pore surface) are 33.7 to 67.4 m^2^/g (CB), and 1.42 to 8.53 m^2^/g (HC), respectively. The densities of both fillers were determined using pycnometry [27] and were also found to be highly different: 2.96 g/mL for CB vs. 1.407 g/mL for HC. This means that the volume fraction of the combined filler phases is significantly increasing when going from VCB50 (filled exclusively by CB: 50 phr) to VCB00 (filled exclusively by HC: 50 phr), namely from 12 to 25 % *v*/*v* (see Table 2 in the Experimental Part, Section 2.3). Additionally, the HC filler was found [27] to preserve its organic character and possess a carbon content of 71 wt.% (somewhat lower than in benzene). The latter properties suggest that HC might exhibit interesting chemical reactivity, which was observed in the present study.

In Figure 1a, the morphology of the vulcanized filler-free ‘neat’ matrix is depicted, while one of the **vulcanized composites** filled with 50 phr of the combined fillers CB and HC is shown in Figure 1b–e. In the neat matrix (Figure 1a), only occasional ZnO nanoparticles can be observed, as irregular black particles sized 100 to 200 nm, and very few smaller spherical nanoparticles, most likely contamination by CB from the mixer machine. In the composites filled either exclusively with 50 phr of HC (VCB00: Figure 1(b1,b2)) or with 50 phr of CB (VCB50: Figure 1d), it can be observed that the **distribution of the fillers** is relatively even in both cases. However, in VCB50, the filler dispersion is not ideally homogeneous: it displays small-scale (sub-micrometer) fluctuations of distribution density (Figure 1d). In VCB10 (Figure 1c), and much less so in VCB40 (Figure 1d), the fluctuation of the CB distribution is distinct, while large particles of the HC co-filler also are visible. In VCB10, the CB nanoparticles are arranged in a pattern of rather isolated ‘islands’ (somewhat less than 1 µm wide), which are separated by even larger CB-free domains, and the large HC particles are relatively isolated. It further can be observed that the large HC particles do not exhibit dramatic adsorption of CB on their boundaries. In the context of the distribution of CB and HC, it can be noted that the CB-free sample VCB00 (filled exclusively by the large-grained HC filler) contains a fraction of small particles (see Figure 1(b1)). These particles can be attributed to HC fragmentation during the mixing of the rubber recipe because the contamination with CB from the mixer is relatively small, as seen in the example of the neat matrix in Figure 1a. The fragmentation of HC makes a part of the previously internal (pore) surface of this filler accessible for crosslinking and other reactions. The wide distribution of the filler size in VCB00 also is well-visible in the larger-scale image Figure 1(b2). The assignment of the respective filler phases, as well as the elemental EDX analysis of the HC domains, was discussed in detail in the previous work [27].

The **kinetics of vulcanization** at 150 °C of the neat rubber matrix, as well as of the rubber composites VCB00 to VCB50, was characterized using rheological tests on a moving die rheometer (see Figure 2). Figure 2a shows the first 30 min of the process, with well-visible differences in the scorch (induction) time values of the studied samples. Figure 2b plots the time-dependent mechanical resistance (torque) for the whole experiment duration of 2 h.

It can be observed in Figure 2 that the filler-free matrix and the **HC-free** sample VCB50 (filled exclusively with carbon black) display the **steepest and mutually similar vulcanization curves**. However, the matrix displays a smaller initial, and markedly lower maximum resistance, in combination with a much longer scorch (induction) time: matrix: 4.1 min vs. VCB50: 2.2 min. Matrix and VCB50 also display very similar cure times *t*_90_ (where 90% of maximum resistance is achieved) close to 7 min, but the crosslinking process itself is faster in the neat matrix, in view of its markedly longer scorch time.

With an **increasing fraction of HC** (when going from VCB50 to VCB10), the **vulcanization curves become flatter** (especially VCB20 and VCB10), and the **scorch times become shorter**: VCB10 practically displayed none. VCB40 still displays a similarly short cure time (*t*_90_ = 7.5 min) such as VCB50, but VCB30 (8.5 min) and the samples with even **lower CB fractions** display **significantly longer times *t*_90_**: VCB20: ca. 12 min, VCB10: ca. 14 min, and VCB00: ca. 18 min. A similar trend is observed for the time values *t*_max_ at which maximum torque is achieved (at even longer times, degradation reduces the torque again: see Figure 2b): matrix: *t*_max_ = ca. 12 min, VCB50: ca. 15 min, VCB40 and VCB30: ca. 18 min, VCB20: ca. 25 min, and VCB10: ca. 30 min. The sample VCB00 filled exclusively with HC also joins this trend with *t*_max_ = ca. 40 min. The **maximum torques** (mechanical resistance values), which give a measure of crosslinking density, decrease with increasing HC content. VCB10 even displays a lower maximum torque than the neat matrix, while in the case of VCB20, the maximum torque is only slightly higher than the one observed for the matrix.

The **sample VCB00 filled exclusively with hydrochar (HC) displays some anomalies** if compared with the other HC-rich composites. Its **maximum torque** during the vulcanization is **distinctly higher** than that of VCB10 and close to the neat matrix and VCB20. At very long test times (in the degradation region), VCB00 displays higher actual torque values than the matrix, VCB10, and even VCB20, as VCB00 displays a slower degradation tendency than these samples. Additionally, VCB00 displays a distinctly higher initial torque than the neat matrix and VCB10. VCB00 has a similar starting torque value as VCB20 to VCB40. More interestingly, **VCB00 displays the longest scorch time** (4 to 5 min) among the studied composites. Because of the subsequent following slow cure kinetics (increase in resistance), the ‘processability window’ of VCB00 can be considered to be even wider. This is in contrast to VCB10, which displays no scorch time (its mechanical resistance immediately starts to increase). On the other hand, in the region of ‘rapid cure,’ the kinetics curve of VCB00 is somewhat steeper than in the case of VCB10 (see Figure 2a). Interestingly, the **degradation of VCB00 sets on later** and **is less pronounced** than in all the remaining samples.

The above observations indicate a **distinct chemical effect of HC** on the vulcanization chemistry, namely the **slowing-down of the curing step** (flatter curves) and also a longer ‘dynamic region’ where the formation of new crosslinks and the degradation of other ones outbalance each other. The **lower values of achieved maximum torques** in the HC-rich samples can be attributed to the effect of the **lower value of the external surface area of the HC filler** particles. However, in the case of **VCB00**, which is exclusively filled by HC, the **fragmentation of the porous HC particles**, observed as the fine filler fraction in Figure 1(b1), could be responsible for the **increased** (in comparison to VCB10) initial, maximum, and ‘final’ **torque values** (caused by the higher external surface area of the fragmented filler). The distinctly lower initial torque value of VCB10 (and also the observed similar volume fraction of small particles in VCB00 and VCB10, see Figure 1c) might indicate that the fragmentation of HC is less prominent if a much finer co-filler such as **CB** is present during the recipe mixing. The decrease of **scorch times** if going from VCB50 to VCB10 (or from neat matrix to VCB50), as well as the long scorch time of VCB00, indicate an **interplay of several effects**, such as the presence/absence of a reactive filler surface ready for crosslinking reactions (more reactive in case of HC), the changing specific surface area of the co-fillers, the increase of the volume fraction of the co-fillers if going from VCB50 to VCB00, and also the **synergic chemistry** of the combined co-fillers HC/CB. The **anomalous behavior of** the sample **VCB00** was also observed in other **properties, which are strongly influenced by the chemistry** of the fillers, while for other properties, such as hardness, storage modulus, or swelling, **VCB00 displayed a ‘normal’ behavior** as a member of the series Matrix, VCB00–VCB50.

In view of the fact that most of the studied samples achieved maximum torque values in times shorter than **15 min at 150 °C**, these **standard vulcanization conditions** were applied to all the prepared samples in order to ensure identical synthesis- (and thermal) history. In the case of the reference sample VCB00, as well as of VCB10 and VCB20, the so-achieved crosslinking was smaller than the maximally possible one. Some details of the vulcanization rheology of the studied systems (15-min-experiments, without the samples “matrix” and VBC00) were also discussed in the previous work [27].

In the sections below, the mechanical (hardness, tensile properties), thermo-mechanical (DMTA), and swelling properties (study of permanent and physical crosslinking density) of the prepared vulcanized composites, are studied, as well as their chemical stability against thermal and oxidative degradation, and also against deterioration in demanding real use conditions.

### 3.2. Mechanical Properties

#### 3.2.1. Hardness and Tensile Properties

The basic mechanical properties of all the studied rubber and rubber composite samples were characterized using hardness and tensile tests. The results, Shore A hardness, tensile strength, and elongation at break, are presented in Table 3. The stress-strain curves are shown in Figure 3a, while the trend lines of the abovementioned mechanical properties (as a function of composition) are displayed in Figure 3b.

A dominant feature in Figure 3a is the high **extensibility of the non-filled ‘neat’ matrix**: it is ca. 2 times higher than that of the rubber composites. Together with the gradual increase of the slope of the tensile curve, this greatly affects the values of tensile strength and tensile toughness of the neat matrix, which are markedly higher than in the case of the composites.

In Figure 3b (and in Table 3), it can be observed **that the Shore A hardness**, the **tensile strength**, and the **toughness increase with rising carbon black (CB) fraction** if going from VCB00 to VCB50. The trends are practically parallel, but the hardness is least sensitive to the co-fillers ratio (and it does not show the ‘VCB00′ anomaly). The increase is most dramatic between VCB10 and VCB20 in the case of strength and toughness, or between VCB20 and VCB40 in the case of hardness. In relative terms, between VCB00 and VCB50, the hardness increases by 31% (1.31 times), while the tensile strength rises by 130% (2.3 times) and the toughness by 86% (1.86 times). These trends can be attributed to stronger interface interaction (including crosslinking) with the finer filler (CB) due to its higher specific surface area. The less efficient vulcanization in the hydrochar-rich samples, which was discussed further above, also plays an important role, and the 15-min-cure achieves less than full crosslinking in the materials which are richest in HC. The tensile strength and toughness strongly drop if going from a neat matrix to VCB00, as both these magnitudes correlate with extensibility (which also drops). The Shore A hardness (which is not affected by extensibility) does not drop but slightly increases if going from matrix to VCB00 (‘normal’ behavior of VCB00).

The **elongation at break** displays approximately an opposite trend to the above-discussed magnitudes if going from VCB00 to VCB50. It first somewhat increases if going from VCB00 to VCB10 (due to less than perfect vulcanization, especially in VCB00). After that, it steadily decreases due to more efficient crosslinking with CB. In general, the less crosslinked products achieved higher extensibility. Similar to strength and toughness, the extensibility also strongly drops if going from matrix to VCB00, as the reactive fillers, such as HC or CB, generate considerable additional crosslinking (which limits the uncoiling of the macromolecular chains). The extensibility was the highest in the case of VCB10: 1.24 times more than for VCB50. Generally, the **materials VCB30, VCB40, and the hydrochar-free VCB50 are very similar** in many properties, including hardness and strength, but in the case of VCB40 and VCB30, the hydrochar **significantly improves their elongation at break.**

#### 3.2.2. Thermo-Mechanical Properties (DMTA)

The thermo-mechanical properties of the studied natural rubber composites were characterized using dynamic-mechanical thermal analysis (DMTA). The neat vulcanized matrix also was included in the comparison. The temperature dependence of storage modulus (*G*′), loss modulus (*G*″), and the loss factor (tan *δ* = *G*″/*G*′) for the vulcanized samples are presented in Figure 4a, Figure 4b, and Figure 4c, respectively. In Table 4 further below, the values of *G*′, *G*″, and tan *δ* at room temperature (25 °C) can be found (besides other results). Figure 5 shows the dependence of the storage modulus measured at 25 °C on the fillers’ ratio (expressed as carbon black “CB” content). The results generally indicate that **carbon black (CB) generates a distinct increase in the crosslinking density** of the vulcanized samples, **but it also simultaneously causes increased internal friction** (viscosity) due to its high specific surface (offered for physical interactions). **HC causes analogous effects similar to CB but to a smaller extent**. This can be especially well seen by comparing the curves for the samples “matrix,” VCB50, and VCB00 in Figure 4.

In Figure 4, it can be seen that all the (vulcanized) rubber samples with different co-filler ratios, ranging from VCB00 (0 phr of CB + 50 phr HC) up to VCB50 (50 phr of CB, no HC), display very similar DMTA profiles. The same can be said about the neat matrix, which differs from the composites by visibly lower values of *G*′ and *G*″ in the rubbery region. In the case of the composites, the temperature-dependent **storage (*G*′) and loss (*G*″)** moduli (Figure 4a,b) display curves of nearly identical shape, but in the rubbery region (above ca. *T* = −20 °C till the highest temperatures) it can be seen, that the increasing fraction of the fine **nanofiller CB raises the values of both moduli**. The effect on both moduli is similarly strong so that the loss factor tan *δ* (= *G*″/*G*′) does not change significantly in the rubbery region (0 to 180 °C) if the fillers’ ratio is varied (see Figure 4c).

The **effect of the fillers’ ratio on the storage modulus** (trend of *G*′ = f (fraction of CB)) is analyzed in Figure 5. If going from VCB00 to VCB50, *G*′ approximately linearly increases with carbon black content in the filler mixture. The effects can be attributed to the **crosslinking matrix–filler**. The addition of 50 phr of HC (VCB00) raises the *G*′ value at 25 °C by the factor of 1.7 in comparison to the neat matrix (see Figure 5 and Table 4), while 50 phr of CB raises *G*′(25 °C) by a factor of 5.1 (the reinforcing effect of CB hence is three times larger). This difference is comparable to the abovementioned difference in the specific surface area of the co-fillers, as measured by BET (78 vs. 21 m^2^/g, which indicates a 3.7 times larger active surface on CB than on HC). The difference in the external surface area (67 vs. 1.4–8.5 m^2^/g) would suggest a much larger difference in the reinforcing effect. The **comparison of the samples VCB00 and VCB50, suggests that HC is fairly reactive in crosslinking reactions** matrix–filler during the vulcanization. In Figure 4a (*G*′ = f(*T*)), the effect of thermal sample degradation can be observed, which becomes sufficiently rapid (in the timescale of the DMTA experiment) at temperatures above ca. 140 °C.

The **loss moduli** (*G*″) in the studied rubbers **dramatically increased** due to the **incorporation of HC and CB** co-fillers (see Figure 4b and Table 4). This might be attributed to purely physical effects, namely to **weak and reversible interfacial interactions** filler–rubber which causes energy absorption upon deformation, in analogy to the emulsion effect in rheology [32]. A more finely dispersed phase with a higher specific surface means a higher interface area and a stronger ‘emulsion effect’ (higher viscosity). As can be seen in Table 4, the addition of 50 phr of HC raises *G″* (25 °C) by a factor of 21 in comparison to the neat matrix, while 50 phr of CB raises it 118 times, which means that CB generates a 5.7 times stronger effect.

The temperature-dependent **curves of tan *δ*** (see Figure 4c and Table 4) also are fairly similar to each other in their shape, but a distinct trend in the height of the glass transition peak can be observed, namely that it increases more than two times if going from VCB50 filled exclusively with CB to VCB00 (filled exclusively by HC). The peak of the neat matrix is even taller; it is three times higher than that of VCB50, indicating a stronger contribution of elasticity (and a weaker one of plasticity) in VCB50 at *T* = *T*_g_, while the neat matrix conversely displays the highest relative plasticity at *T* = *T*_g_. Secondly, a fillers-ratio-dependent difference can be seen in the intensity of a shoulder on the higher-temperature slope of the glass transition peak. This shoulder can be assigned to polymer matrix chains which are partly immobilized by the proximity of the surface of the rigid filler and which are eventually also immobilized by chemical crosslinking (during vulcanization) between some repeat units of these chains and the filler surface. The superposition of the main *T*_g_ peak and the differently prominent shoulder leads to slight shifts of the glass transition temperature (main peak maximum), namely between the ‘extreme’ values of −52.8 °C (neat matrix) and −49.0 °C (VCB40). It can also be noted that the **tan *δ* values in the rubbery region** (see, e.g., values for *T* = 25 °C in Table 4) are **relatively high** (ca. 0.1) compared with ideal rubbery polymers (0.003–0.005, see [32]) and also with the neat matrix (tan *δ*(25 °C) = 0.007, see Table 4). The mentioned high tan *δ* values are characteristic of commercial, highly filled rubber composites typically used for tire production.

### 3.3. Swelling of the Hydrochar Nanocomposites

In order to **evaluate (permanent and reversible) crosslinking in the studied rubber composites** and in the neat matrix, 24-h-swelling experiments in a toluene bath were carried out at room temperature (25 °C), the solvent uptake kinetics were recorded, and also the dynamic-mechanical properties (moduli *G*′ and *G*″, as well as tan *δ*) of swollen dry and specimens (at 25 °C) were compared. Swelling equilibrium was reached well before 24 h in all cases. The curves of the time-dependent swelling degree (= 100% × Δ*m*/*m*_initial_) of the vulcanized nanocomposites are depicted in Figure 6. The measured dynamic-mechanical properties of dry and swollen samples are compared in Table 4. Some calculated magnitudes concerned with the concentration of elastically active polymer chains (‘crosslinking’ density) also are listed in the table.

Generally, the swelling-related experiments indicate that the strongest **permanent crosslinking** is present in the sample reinforced exclusively by CB, namely VCB50, and that the crosslinking **increases with increasing CB fraction** in the co-fillers phase. Additionally, a **prominent role of reversible crosslinking** (entanglements and filler–polymer interactions) in all the non-swollen composite samples is revealed. The reversible crosslinking (which seems to be nearly completely disconnected in the swollen state) is generated by both co-fillers, but the effect of CB is stronger.

The **swelling (solvent uptake) curves** of all the studied samples are shown in Figure 6. They display very similar shapes and indicate a ‘parallel’ course of all of the swelling kinetics, but the curves visibly differ in the achieved maximum swelling degree (100% × Δ*m*/*m*_initial_). The neat matrix displays the highest equilibrium swelling degree among the studied samples, namely 346% (corresponding to a swelling ratio *Q* = *m*_swollen_/*m*_initial_ = 4.46 in Table 4, where *Q* = (*swelling_degree* + 100%)/100%). In contrast, the sample which is filled exclusively with CB, VCB50 (and which displays the highest specific surface area of the filler phase), conversely reaches the lowest equilibrium swelling degree of 211% (*Q* = 3.11). The results in Figure 6 indicate a **steady increase in permanent (covalent) crosslinking density** if going **from** the neat **matrix** through VCB00 **to** the sample **VCB50**. This conclusion also was hinted at by the tensile, hardness, and DMTA tests discussed further above. These latter tests, however, were sensitive to the combined, permanent + reversible crosslinking density.

If the **course of the swelling kinetics** in Figure 6 is considered in detail, it can be seen that the most crosslinked (and hence lowest-swelling) sample VCB50 displays the fastest kinetics and reaches equilibrium in ca. 4 h. The highest-swelling ones, the neat matrix, VCB00, or VCB10, need ca. 6 h. Most of the swelling occurs in ca. 3 h, while the most dramatic changes occur in 1 h (VCB50) to 2 h (matrix). This trend seems to correlate with the relative amount of solvent, which must be taken up via diffusion, and not with reversible physical crosslinks, where their effects are discussed below.

The **analysis of** the trends in **magnitudes** obtained **from dynamic-mechanical analysis** of dry and **swollen samples** (see data in Table 4) yields detailed conclusions about **reversible vs. permanent crosslinking** and internal friction (viscosity) in the studied rubber nanocomposites. A big difference between their dry and swollen state is that the relative viscosity/plasticity (expressed as the loss factor tan *δ*) dramatically decreases if they undergo swelling: tan *δ* drops from ca. 0.1 (dry) to 0.003 (swollen) so that the swollen rubber nanocomposites correspond to ideally rubbery polymers in their viscoelastic properties [32] (in contrast to the dry ones). The filler-free matrix is very different in this respect: it displays a tan δ value close to ‘ideal’ elastomers, and the drop in its value upon swelling is much less dramatic than in the case of the composites (from 0.007 down to 0.001). The reason for the change in the viscoelastic character of the composites is that as a consequence of the equilibrium swelling, *G*″ decreases much more dramatically (by more than two orders) than does *G*′ (by less than one order). In the dry, rubbery state (at 25 °C), the value of *G*″(25 °C) can be observed to markedly increase with the increasing fraction of CB in the fillers’ mixture, similarly markedly as *G*′. This was already noted further above (DMTA analysis). In the case of the swollen samples, *G*″ is always fairly small and only slightly rises with increasing CB fraction in the co-fillers phase. The **dramatic decrease of *G*″ after completed swelling can be assigned to the disconnection of easily reversible crosslinks** (entanglements and/or physical filler–polymer interactions) due to polymer-solvent interactions. The modest dependence of *G*″ on the co-fillers ratio in the swollen samples seems to reflect the influence of the specific surface area of the co-fillers phase. In the filler-free matrix, the value of *G*″ is dramatically (20 to 120 times) smaller than in the composites if dry samples are compared, while it is several times smaller if swollen ones are considered. This finding highlights the prominence of strong (effect on *G*′ in a dry state) and weak (effect on *G*″ in a dry state) reversible physical crosslinks of the matrix–filler type in the composite samples.

In order to evaluate permanent vs. reversible crosslinking in more detail, nominal **concentrations of elastically active polymer chains** (*c*(*EAC*) ≡ ‘crosslinking density’) can be **calculated** using the idealized formula [32]:$$G=\frac{c\left(EAC\right)}{RT}$$
where *G* is the shear modulus (here, the experimental *G*′), *R* is the universal gas constant, and *T* is the temperature (here 298.15 K). The *c*(*EAC*) values calculated for all samples from experimental G′ values are listed in Table 4. Additionally, in the case of the swollen samples, the theoretically expected *c*(*EAC*)_SW,theor._ values (based on *c*(*EAC*)_non-swollen_ and on the swelling ratio *Q*_equilibrium_) were also calculated using the above formula and the assumption that the molar amount of the elastically active polymer chains (*n*(*EAC*)) should remain unchanged, while the sample volume increases by the factor equal to the swelling ratio *Q*. Hence the following formula was used:$$c{\left(EAC\right)}_{\mathrm{SW},\mathrm{theor}.\;}=\frac{c{\left(EAC\right)}_{\mathrm{non}\_\mathrm{swollen}\;}}{Q}$$

The comparison of the theoretical *c*(*EAC*)_SW,theor._ with the experimental *c*(*EAC*) values of the **swollen samples** (as factor *c*_sw_/*c*_sw,theor._ in Table 4) was used to verify whether the molar amount of crosslinks *n*(*EAC*) is constant. It can be seen from Table 4 that the theoretical values *c*(*EAC*)_SW,theor._ are 2 to 3 times higher than the experimental *c*(*EAC*) values of the swollen rubber composites. This indicates the presence of a **considerable fraction of reversible but strong and elastically active** (in a dry state) **crosslinks in the studied materials**, which are ‘dissolved’ during the swelling, as also suggested by the above-discussed dramatic change in tan *δ*. The factor *c*(*EAC*)_SW_/*c*(*EAC*)_SW,theor._ (abbreviated as “*c*_sw_/*c*_sw,theor._” in Table 4) generally moderately decreases with increasing CB fraction in the co-fillers phase. This indicates an increasing fraction of elastically active but reversible crosslinks at higher CB contents. The effect might be attributed to the higher specific filler area of the fine-grained component CB. However, the trend is not simple: A local minimum is at the CB concentration of 30 phr (combined with 20 phr of HC). This result might be caused by the chemical effect of hydrochar, which influences the crosslinking and hence also the probability of the formation of entanglements or physical crosslinks. In the case of the neat matrix, the value of *c*_sw_/*c*_sw,theor._ is greater than 1 (namely 1.684, see Table 4); hence, *G*′ is distinctly less reduced upon the swelling of the neat matrix than in the composites (and also in comparison with simple theory). This could be attributed to permanent entanglements in the filler-free network. This type of entanglement, which increases the modulus *G*′ in the swollen state, was not taken into account in the case of the *c*_sw_/*c*_sw,theor._ ratios calculated for the composites (a correction would reduce the estimated fraction of covalent crosslinks). Using the values of *c*_sw_/*c*_sw,theor._ from Table 4, the ‘permanent components’ of the storage modulus *G*′_permanent_, as well as of the related ‘density of permanent crosslinks’ *c*(*EAC*)_permanent_ were estimated for the dry composites, using the formulas *G*′_perm_. = *G*′ × (*c_sw_/c_sw,theor_*_._) and *c*(*EAC*)_perm._ = *c*(*EAC*) × (*c_sw_/c_sw,theor_*_._). A relatively simple trend can be seen, namely an increase in *G*′_perm_. and *c*(*EAC*)_perm._, if going from the neat matrix via VCB00 to VCB50. The trend could be attributed to additional covalent filler–matrix crosslinking, made possible by the increasing specific surface area of the co-filler phase. *G*′_perm_. of VCB00 is slightly lower than the *G*′_perm_. of the neat matrix. This anomaly is most probably connected with the imperfect cure of VCB00 at the standard time of vulcanization (which was discussed further above in Section 3.1).

### 3.4. Chemical Stability

#### 3.4.1. Thermal Stability: TGA

The **stability** of the prepared vulcanized rubber composites **against oxidation**, as well as against **anaerobic thermolysis,** was investigated using **thermogravimetric analysis** in air and in nitrogen, respectively, at the heating rate of 10 °C/min. The temperature dependence of the relative sample mass (‘TGA proper’) and of its derivative (dTGA) are presented in Figure 7 and Figure 8, respectively, for both types of TGA tests (in air and N_2_). It can be summarized that the variation of the CB/HC ratio in the fillers phase only has a **modest effect on the temperatures of** the onset and maximal rate of **decomposition**, both in air and in nitrogen. Nevertheless, there appears to be a **detectable synergic stabilizing effect** of the combined fillers, which in air is maximal at 10 and 20 phr of HC (VCB40 and VCB30), and in nitrogen at 20 and 30 phr of HC (VCB30 and VCB20), respectively. Even stronger is the anti-oxidizing effect of HC if it is used as the exclusive filler in VCB00. This composite has the highest temperatures of maximum decomposition (peaks in Figure 8), both in air and nitrogen. Even more importantly, the intermediate char fraction (forming above 400 °C), which is observed in VCB00, is distinctly higher than in VCB10 and approximately equal to the one in VCB20. In nitrogen, the permanent char residue of VCB00 also is higher than would be expected from the trends. Apart from the **anomalous behavior of VCB00**, a distinct effect of the **increasing HC fraction** is the **decrease of** the temporary (in air) or permanent (in nitrogen) **char fraction**. The strong stabilizing effect of HC in VCB00, as well as the significant synergic stabilization in VCB20–VCB40, can be attributed to the **reactivity of HC** (to its abovementioned organic character, characterized in detail in [27]), which later, at the highest temperatures, leads to faster oxidation or pyrolysis of the previously stabilized char fraction. Interestingly, the chemical interaction of HC with CB leads to a less efficient stabilization in VCB20, especially in VCB10. Details of the observed TGA trends are discussed below.

**The course of the weight loss curves**: the **degradation** process in both atmospheres **always starts at about 200 °C** in all the samples (see Figure 7a,b), while **5% of weight loss is reached between 310 and 350 °C.** Near **390–400 °C**, all the studied samples in both atmospheres **rapidly lose** a substantial amount of their mass. This ‘step’ in the curves corresponds to the nearly complete **degradation of the polymeric matrix** (natural rubber). Because of nearly identical (at the same phr of CB) decomposition temperatures as well as curve courses in both air and nitrogen, the degradation seems to have an analogous first mechanistic step, both in air and in N_2_, namely rubber thermolysis, rather than oxidation (which follows as a subsequent process in air). Distinct differences in the TGA curves are noted at higher temperatures; **above 420 °C**, a temporary (in air) or permanent (in nitrogen) **char fraction** is observed, formed via thermal fragmentation, followed by oxidative or anaerobic crosslinking. In both atmospheres, the char fraction is the highest if the filler phase consists exclusively of carbon black (sample VCB50), and it decreases with an increasing fraction of HC in the fillers phase (due to the organic character of HC). However, as mentioned above, in the case of VCB00 (filled exclusively by 50 phr of HC), the char step is anomalously high, which suggests a strong anti-oxidizing and radical reactivity of HC (the effect is moderated by the addition of small amounts of CB, although the latter also has a stabilizing effect).

**In air**, the **oxidative degradation of temporarily formed char** becomes fast near 650 °C. Above 700 °C, all organic residues are degraded in the air atmosphere. There **remains** an **ash residue of ca. 3 to 4%,** which does not undergo any changes until 800 °C (see Appendix A). This small but measurable ash fraction can be attributed to zinc oxide (ZnO) which was added to the rubbers (4 phr corresponding to 2.5 wt.% in the recipe in Table 1). Its eventual partial reaction with oxidized sulfur (partial transformation into ZnSO_4_) and also the small contribution of mineral ash from hydrochar (0.42 wt.% in the HC filler, as determined in previous work [27]) could be responsible for the higher ash content, than that corresponding to pure ZnO from the rubber recipe.

In **nitrogen**, the **char content reaches final values** already near 550 °C in most samples (finished carbonization). The **final char residues** at 800 °C are **between 19 and 34 wt.%** (see Figure 7b, except the filler-free matrix, where the residue is just 5.1 wt.%, slightly more than the ZnO content), depending on the CB/HC ratio (see Appendix A). For comparison, the content of the combined filler phases was always equal to 31.45 wt.% (≡50 phr in the rubber recipe in Table 1 in Section 2. Materials and Methods), to which the CB component contributed by 0 to 31.45 wt.%, and HC by 31.45 wt.% down to 0 wt.%. In the case of VCB50 (see Appendix A), only 3.2 wt.% in the final char (34.65 wt.%) originates from components other than the original amount of CB (31.45 wt.%). These 3.2 wt.% well correspond to the mentioned content of ZnO in the rubber recipe in Table 1 (2.52 wt.%; thus, the additional residue other than ZnO and CB amounted to just 0.68 wt.%). Hence, the thermolysis of natural rubber to gaseous products can be considered practically quantitative in nitrogen at T > 450 °C; if hydrochar (HC) is incorporated, the final char residue increases (see Figure 7b), but less so than would correspond to the content of the HC filler. According to calculations in Appendix A (subtraction of inert CB and ZnO), an additional residue ranging between 3.93 wt.% (in VCB40) and 16.78 wt.% (in VCB00) can be attributed to HC, whose content in the intact sample was 6.29 to 31.45 wt.%, however. Hence, **only 53–62** wt.% **of the hydrochar mass survived the thermolysis in nitrogen** as a consequence of the organic character of HC (its carbon content was only 71%, as found in [27]). Due to the mentioned anomaly of VCB00 (very strong anti-oxidizing properties of pure HC), the smallest carbonized fraction of HC was found in VCB10 (46% of the original mass).

Figure 8 shows the **derivative analysis (dTGA) of the above TGA results**. The negative peaks of the “relative weight derivative” indicate temperature points at which degradation rates achieve local maxima (rapid weight losses) and signalize different decomposition processes.

**Both in air and in nitrogen**, the abovementioned **dominant degradation process**, the **thermolysis of** the **natural rubber** polymer, is signalized by the intense dTGA peak near ***T* = 390 °C**. This process corresponds to the main step in the TGA curves in Figure 7. The main degradation peaks of all the studied samples, both in air and in nitrogen, display very similar positions and also nearly identical intensities. In air, the peak slightly shifts from 391.5 °C to 397, 383, to 390, 394, 394, and then back to 388 °C if going from the neat matrix via VCB00 to VCB50. This trend in the temperatures of decomposition maxima confirms the abovementioned **strong anti-oxidizing effect of CB-free HC, as well as the synergic stabilizing effect of HC and CB at small to medium HC concentrations** (VCB40, VCB30, and VCB20). The low temperature of the main peak of VCB10 (*T*_dec_ = 383 °C) indicates that at this co-filler ratio, the chemical effect of HC is being moderated by the CB co-filler, as already suggested by the mechanical analysis of the swollen samples (effect on crosslinking during vulcanization). In the nitrogen atmosphere, the trends of the main decomposition peak are nearly identical to those in air, including the shifting temperatures of the decomposition maxima, namely 396.5, 399.5, 386, 397, 400, 393, and 386 °C, if going from the neat matrix via VCB00 to VCB50. VCB10 displays an analogous instability in nitrogen to that in air. The above results suggest that **hydrochar (HC) exerts an analogous chemical influence on rubber thermolysis during aerobic and anaerobic degradation**. Finally, it can be noted that the neat matrix displays the most intense main decomposition peak, both in air and in nitrogen. This is because this sample contains only one significant component, the natural rubber itself.

**Separate oxidation of the co-fillers during degradation in air:** In the case of the **dTGA curves** of samples tested in air (see Figure 8a), two smaller peaks can be observed in the temperature region from 400 to 700 °C. They correspond to the structured smaller step in the mass loss curves (see Figure 7a) in the same temperature region. The peaks near 520 and 640 °C were assigned to the oxidation of hydrochar (together with an eventual charred fraction of the matrix) and of carbon black, respectively, as will be explained below.

**CB oxidation (*T*_max_ = ca. 640 °C):** The height of the second (final) TGA sub-step in Figure 7a, ranging from 550 to 700 °C (corresponding to the dTGA peak at 640 °C in Figure 8a), which is part of the mentioned structured TGA mass loss step extending from 400 to 700 °C in Figure 7a, clearly correlates with the CB content. The sum of the weight percentage of embedded CB plus the percentage of embedded ZnO well corresponds to the height of the mentioned 550–700 °C sub-step (see Appendix A). In the example of the HC-free composite VCB50, it can be seen that the height of the 550–700 °C sub-step (here, the first, lower-temperature sub-step is practically absent) in Figure 7a is ca. 35.0%, while the sum of the contents of ZnO + CB in that sample is 33.97%. Similar observations can be made for the other nanocomposites as well (see Appendix A: e.g., VCB10: ca. 10.5% vs. 8.81%). The intensity of the dTGA peak associated with the oxidation of the CB filler increases with the CB content. Its position moderately shifts, as it is influenced by the previous oxidation of the hydrochar and charred matrix. If going from VCB10 to VCB50, the peak moves from 627 °C to 645, 656, 678, and finally slightly back to 669 °C.

**Oxidation of Hydrochar + Charred Matrix (*T*_max_ = ca. 520 °C):** The ‘first’ TGA sub-step (region: 400–550 °C in Figure 7a, corresponding to the peak at 520 °C in dTGA in Figure 8a), which is part of the discussed structured TGA mass loss step at 400–700 °C in Figure 7a, logically was assigned to the simultaneous oxidation of the charred matrix and of hydrochar, as the latter filler is less resistant to oxidation than CB, due to the organic nature of HC. The small but significant exclusive **contribution of the charred matrix (via oxidative crosslinking)** is visible both in Figure 7a (steps) and in Figure 8a (small dTGA peak) in the case of VCB50, which contains no HC at all, but still displays a small ‘first sub-step’ (height ca. 7%; combined the 1st and 2nd steps: ca. 42%). In all samples, the combined height of the sub-steps at 400–550 °C and 550–700 °C roughly corresponds to the combined fractions of ZnO, CB, and hydrochar (see Appendix A). The contribution of the 1st sub-step to the combined structured step expectedly increases with the fraction of the embedded HC. In VCB00 (with 50 phr of HC, where the second sub-step is absent), the height of the first sub-step is ca. 30.5%, which is slightly less than the HC content of 31.45%. However, at low hydrochar contents (e.g., in the mentioned VCB50 and VCB40), the 1st step is somewhat higher than the HC content due to the presence of a charred matrix, whose content is evaluated in the last column of Appendix A. At high hydrochar contents (VCB00, VCB10, and less so VCB20), on the other hand, the 400–550 °C sub-step is lower than expected (negative values for charred matrix fraction in Appendix A), which means that **a part of the hydrochar degrades during the degradation of the natural rubber.** The small dTGA peak near 520 °C, which corresponds to the simultaneous oxidation of hydrochar and the charred matrix, steadily decreases with decreasing content of hydrochar (the associated sub-step in Figure 7a diminishes). The position of this dTGA peak shifts from 513 °C to 526, 534, 539, and back to 525 °C if going from VCB10 to VCB50. As noted in other chemistry-related properties, VCB00 (with HC as the sole filler) displays an anomalous anti-oxidizing stabilization. The peak of the oxidation of HC and the charred matrix is shifted to markedly higher temperatures, namely to 585 °C (between the dTGA peaks of the first and second sub-step). As also mentioned above (and as observed in other chemistry-related properties), in VCB40, which is close to VCB00 in its composition, the stabilizing effect of HC and CB is mutually canceled. Finally, the neat matrix also generates an intermediate char fraction upon oxidation (see Figure 7a: nominal step height including ash 9.1%, char itself: 4.35%), whose decomposition maximum is somewhat higher than even that of VCB40, namely at 553 °C. The shifting degradation maxima of CB, as well as of HC + the charred matrix (near *T*_max_ = ca. 640 °C and *T*_max_ = ca. 520 °C, respectively), further confirm the finding **that CB-free HC displays a strong anti-oxidizing effect where moderate to medium contents of hydrochar (HC) in combination with CB exert a synergic chemical stabilizing effect** and that the mentioned stabilization affects **both the oxidative and the anaerobic degradation** processes.

#### 3.4.2. Thermo-Oxidative Aging Test

The above-discussed analyses at elevated temperatures indicated that rapid thermal and/or oxidative degradation onsets above 200 °C (in the fast TGA analyses performed at 10 °C/min) or even near 150 °C (in DMTA analyses at 3 °C/min). Hence, the moderately elevated constant temperature of 180 °C was selected for performing thermo-oxidative aging tests ‘at harsh conditions’ by exposing the prepared samples to circulating air at this temperature (and in darkness) for different periods of time. Milder aging tests simulating real-use conditions were performed (see further below 3.4.3 Simulated weathering).

In the design of the thermo-oxidative aging test, the authors’ previous experience with the study of oxidation stability of poly(propylene oxide)-based polymer networks [31] was useful, where differences between the samples led to marked effects in relatively short treatment times. The thermo-oxidative aging tests were performed using an oven with forced air circulation. Several specimens of each tested material were prepared. Each specimen was then subjected to a specific oxidation time at 180 °C, namely to 30, 60, 180, or 360 min. The mass of each specimen was recorded before and after the oxidation treatment. The **oxidation-time-dependent relative masses** of all the tested materials are presented in Figure 9 (the experimental data can be found in Table 3). Each oxidized specimen was subsequently subjected to thermo-mechanical analysis (DMTA) to evaluate oxidation-time-dependent changes in its properties (the results of these analyses are discussed further below).

As seen in Figure 9 (and in Table 3), the **mass losses** of the tested specimens during the thermo-oxidative aging tests in the oven **were relatively modest** (in spite of the released odor). After 6 h of these tests, the final mass losses ranged between 1.5% and 3.5% (Figure 9), although an error margin of at least ±0.5% must be considered (because of manipulation with the samples). Due to the error margin, the small oscillations seen in the zoomed Figure 9, Figure 10, Figure 11 and Figure 12 should not be considered local trend reversals. Moreover, the TGA analysis did not indicate net mass uptake in any temperature region, e.g., via oxygen incorporation. If the trends (within error margins) are compared, it can be stated that the onset of oxidation-induced mass loss (in the first 30 min) is accelerated by the presence of hydrochar (HC), but later (3 h, 6 h) the mass loss practically stops in HC presence, while in its absence it continues (see trendline of VCB50 in Figure 9). These results again confirm the **chemical effect of the hydrochar,** which **accelerates the onset of degradation** on the one hand (see the first 30 min; possibly due to the anti-oxidizing properties of HC, which could favor unhindered thermolysis) but later **stabilizes the composites** (slowed-down mass loss in later stages in the samples VCB30, VCB40). An approximate trend (while still considering the significant error margin) can be observed. The smallest and slowest mass loss occurred in VCB50, while the largest and fastest one was in VCB00. The neat matrix itself degrades slightly more and also initially faster than VCB50. This trend is opposed to the trend in the below-discussed constancy of mechanical properties, where despite the mass loss trends, the sample VCB00 displays the distinctly highest constancy (the stabilizing effect of HC hence is connected with mass consumption).

The **dynamic-mechanical thermal analysis (DMTA) of oxidized specimens** yielded temperature-dependent plots of the storage modulus (*G*′), of the loss modulus (*G*″), and of the loss factor tan *δ*. The results are analyzed in Figure 10, Figure 11 and Appendix A and in Appendix A: Overlaid DMTA curves of selected samples can be seen in Appendix A, in Figure 11, and in Appendix A. Oxidation times, after which the different samples are compared in the latter figures, are 30, 60, 180, and 360 min, respectively. Additionally, in Figure 12, the effects of the progressing thermo-oxidative aging on the examples of selected samples are compared.

**General trends:** After oxidation times of 30 and 60 min, all the DMTA curves displayed practically identical shapes as in the case of the intact samples prior to oxidation, which can be seen in Figure 4 further above (Section 3.2.2. Thermo-mechanical properties (DMTA)). The only difference between the intact, 30-min, and 60-min oxidized samples was in the values of their moduli in the rubbery region. After 180 min of the thermo-oxidative aging test, the DMTA curves of the rubber composites are already markedly altered, as can be seen in Figure 11 (similar curves after 360 min are shown in Appendix A). The above Figure 10 plots the **trends of the changes in dynamic-mechanical properties** measured at room temperature, namely in *G*′ (25 °C), *G*″ (25 °C), and tan *δ* (25 °C), **as a function of the oxidation time.**

The **changes in the storage modulus *G*′ (25 °C)**, which result **from the endured time of the thermo-oxidative aging test** (Figure 10a), supply interesting information about the crosslinking density in the aging samples. It can be seen that *G*′ (and hence the crosslinking density) initially significantly decreases during the first 30 min, after which (at 60 min) the modulus (crosslinking density) either stagnates (matrix, VCB00, and VCB10), or it increases again (VCB40, VCB50). Next, a steady modulus increase is observed for most samples, but in VCB40, which together with VCB30 often was noted as displaying a synergic stabilizing effect of the co-fillers, the modulus increases only between 30 and 180 min, and after that (180–360 min) it stagnates or even slightly decreases. The trends of *G′* suggest that two processes occur during the thermo-oxidation tests. **At first (0 to 30 min), due to the presence of anti-oxidants** in the commercial recipe of the studied rubbers, a slow **quasi-anaerobic thermolysis occurs.** Oxygen is at least partly captured, and radical reactions are blocked by the anti-oxidants, while the polymer matrix network thermally degrades (polysulfide crosslinks can start to dissociate). **After 30 (or 60 min)**, the anti-oxidant additives are consumed, and the reactions between natural rubber and oxygen prevail and lead to **additional oxidative crosslinking**, which causes the modulus to rise (the material becomes stiffer and becomes brittle). This oxidative crosslinking is the fastest in hydrochar-free VCB50 and in the neat matrix, while it is distinctly slower in the HC-rich samples VCB10 and VCB00 (see the smaller slope of the curve in Figure 10a, between 60 and 360 min). The sample VCB40, which contains only 10 phr of HC but in which a stabilizing synergy effect of the combined filler pair was observed by several other methods, displays initially (from 0 to ca. 120 min) the same behavior with VCB50, but after 180 min, the oxidative crosslinking in VCB40 is wholly suppressed (slightly decreasing modulus instead of steady growth). This also correlates with the practically stopped mass loss of VCB40 after 180 min of the thermo-oxidative aging test (as discussed above, Figure 9b).

**Trends of *G*″ (25 °C) as a function of the oxidation time** are very similar to the trends of *G*′ (see Figure 10b vs. a), but the range of values of *G*″ is narrower if the behavior of the neat matrix is not considered. In the initial oxidation period (presence of the anti-oxidant additives), between 0 and 30 (60) min, *G*″ just stagnates or only very slightly decreases in contrast to the strong decrease in *G′*. In later stages, *G*″ moderately increases in all samples except VCB40 (which displays synergic stabilization). The lowest *G*″ values among the composites are observed for VCB00 and VCB10. The neat matrix, in contrast to the composites, displays a nearly steady increase of *G*″ with the duration of the thermo-oxidative aging test. This increase markedly slows down after 3 h. The factors responsible for the *G*″ trends are similar to the case of *G*′. In the early stage, the thermolysis of crosslinks does not significantly influence the molecular friction responsible for *G*″ in the case of the composites crosslinked by numerous bonds to the filler phase. Without the filler, however, the thermolysis generates an immediate increase of *G*″ in the matrix. In the later stage of the thermo-oxidative aging test, from 30 (60) to 360 min, the oxidative crosslinking reactions produce not only network junctions but also branched structures in all the samples. The branched structures then raise molecular friction and hence *G*″.

**Trends of tan *δ* (25 °C) as a function of the oxidation time** (see Figure 10c) follow from the trends of *G*′ and *G*″, as well as from the precise values of both. Here it can be seen that in the **initial period** (0 to 30 (60) min), the **anaerobic degradation markedly raises tan *δ*** in all samples, while the **subsequent period of oxidative crosslinking** practically leads to a **stagnation** of the tan *δ* value (VCB40), or to its modest (VCB10) or even distinct **further increase** (VCB00). In VCB50 and in the neat matrix, where the additional crosslinking is very prominent in the late stages (see Figure 10a), tan *δ* markedly drops between 180 and 360 min, approximately to (VCB50), or close to (neat matrix) its starting value.

**The glass transition temperatures (*T*_g_),** obtained from the tan *δ* = f(*T*) graphs in Appendix A, Figure 11c and Appendix A, were defined here as temperatures of maxima of the tan*δ* peaks in the mentioned graphs. In contrast to *G*′ and *G*″, the tan *δ* = f(*T*) graphs are much more sensitive indicators of endured oxidation and of the resulting structural changes. The intact rubber composites display very simple single tan δ peaks at their *T*_g_ (see Figure 4c further above). A special case is the intact VCB50, where an overlaid peak of an immobilized matrix fraction can be recognized as an asymmetry in the main transition peak, including a shoulder on the higher-temperature slope of the main *T*_g_ peak. Nevertheless, the *T*_g_ values of the intact rubber composites are practically identical.

After **30 min of the thermo-oxidative aging** (Appendix A), the ***T*_g_ peaks** are still very similar to that in the intact samples (in Figure 4c), but the peak heights are significantly lower, and the values of tan δ in the post-*T*_g_ plateau are higher. In the case of VCB50, the overlaid peak of the immobilized matrix nearly becomes quite visible. In the other samples, the symmetry of the peaks slightly varies (overlaid transitions) so that the nominal *T*_g_ values oscillate. Additionally, the nominal *T*_g_ values of all samples are slightly up-shifted (by 2 to 5 °C) compared to the intact state, indicating a somewhat more prominent role of the fraction of immobilized matrix.

**After 60 min of thermo-oxidative aging**, its effects on the ***T*_g_ peaks** become very distinct. The overlaid peak of the **immobilized fraction** in VCB50 **becomes much more prominent** (and broader). In VCB40, the transitions of free and immobilized matrix fractions are not yet separated, but their superposition yields a distinctly broadened and unsymmetrical single peak with a highly up-shifted (in *T*) main maximum (intact: −48.99 °C after 60 min: –29.35 °C). In VCB10, filled with 40 phr of HC (+ 10 phr of CB), the changes in the main transition peak are still modest after 60 min, but the nominal *T*_g_ value is up-shifted by several additional degrees (due to the immobilized fraction).

**After 180 min of the thermo-oxidative aging test**, the ***T*_g_ peaks** in Figure 11c are very different from the ones of the intact samples. **Separate transition peaks** are observed for the **free matrix fraction** and for the prominent **immobilized fraction,** even in the case of VCB10 (in which sample the peaks still partly overlap). The separate *T*_g_ peaks indicate **micro-phase separation between domains of the free and immobilized matrix**. The transition of the free matrix is slightly down-shifted (in *T*) in VCB50 and VCB40 in comparison to intact samples (due to more irregular structure), while in VCB10, it is slightly up-shifted (in *T*), but less so than after 60 or 30 min. The prominent immobilized phase displays relatively high *T*_g_ values: −10 °C (VCB10), +15 °C (VCB20), +40 °C (VCB30), +23 °C (VCB40), and +27 °C (VCB50), respectively. The values of tan *δ* in the post-*T*_g_ plateau are relatively high. The **oxidized matrix** displays the highest *T*_g_ of the immobilized fraction (+32 °C), and the peak of this fraction is more prominent than the one of the intact matrix fraction (which is below the analogous peak of VCB50 in Figure 11c). Hence, the **crosslinking in the neat matrix is most prominent**. On the other hand, **VCB00**, after 180 min, displays just a shoulder (centered on 0 °C and extending from ca. –20 to ca. +20 °C) on the broadened main transition peak, which indicates the **slowest progress of oxidative crosslinking** in this sample. In VCB10, where the tan *δ* = f(*T*) curve displays a similar course to the one of VCB00, the peak of the immobilized fraction is already separated from the main transition.

**After 360 min (6 h) of the thermo-oxidative aging** (Appendix A) the changes in the *T*_g_ peaks intensify. The peaks of the immobilized phase get broader and shift/extend to even higher temperatures. Except in the HC-free sample VCB50, the *T*_g_ peak of the ‘free’ matrix also broadens and becomes structured (indicating further chemical changes of the original ‘free matrix’).

The *G*′ = f(*T*) and *G*″ = f(*T*) **curves of the strongly oxidized samples** (see Figure 11a,b: after 180 min, and Appendix A: after 360 min, as well as Figure 12: general trends in the curves of *G′*) display **distinctly different shapes** if compared with the curves of the intact samples in Figure 4a,b. In the case of both *G*′ and *G″*, an additional step appears at temperatures above the main glass transition (while the main *T*_g_ is marked by the initial (low-temperature-) step in *G*′ and by a peak in *G*″). The step additional high-temperature step in *G*′ is small in VCB10 and undistinguishable in VCB00 (here, the slope of the main step becomes less steep). On the other hand, the additional step is most prominent in the neat matrix (where the cracking of the specimen can be observed near 150 °C in Figure 11c).

The **additional step in *G*′** results from **oxidative crosslinking, which yields an immobilized phase**, while it also yields **branching and, hence, additional friction** (smaller steps in *G″* in Figure 11a,b and Appendix A). Interestingly, in the case of the sample VCB40, for which a stabilizing synergy effect of the co-fillers was observed in several different experiments, the additional steps in *G*′ and *G*″ decrease (simultaneously with the stopped weight loss) if the oxidation time is increased from 180 min to 360 min (see Figure 11a,b and Appendix A). **In the case of VCB40**, the chemical **effect of hydrochar** in synergy with CB **seems to partly reduce the micro-phase separation** caused by oxidation.

Figure 12 focuses on **the changes in the *G*′ = f(*T*) curves** which appear **with increasing duration of the thermo-oxidative aging test, by comparing the development in selected samples**. It can be observed that the **oxidative crosslinking** is fairly **efficiently blocked** (no second step in *G*′) **in the sample VCB00**, filled exclusively by HC, especially in the first 3 h. Crosslinking in VB00 sets on between 3 and 6 h, but no second *G*′ step appears even after 6 h. In **VCB40**, where a **synergy effect** of the fillers was observed already with other characterization methods, the formation of the second step in G′ is much less prominent than in VCB50 (filled exclusively by CB) and compared to the neat matrix. These findings further demonstrate the **stabilizing chemical effects of HC and HC/CB**.

#### 3.4.3. Simulated Weathering: Industrial ‘Florida Test’

The **weathering** of the prepared rubber composites **in real-use conditions** (less harsh than the thermo-oxidative aging test at 180 °C) was tested as “accelerated aging” according to the method PV 3930 (‘Florida Test’) developed by Volkswagen, Germany, which relies on the guideline ISO 4892-2. The Florida Test (setup: see Experimental Part) simulates a demanding, hot and humid climate, where the sunny weather is simulated by UV irradiation of the wavelength of 340 nm. One real ‘Florida day’ is simulated by a 120-min-treatment so that in one day (24 h) of the uninterrupted aging test, 12 real days are simulated. The results of the weathering tests are summarized in Figure 13, Appendix A.

Selected **samples**, namely **VCB40, VCB10**, and **VCB50**, were subjected to two differently long **aging** treatments: **7 days** (corresponding to 2.75 real months in Florida, USA) **and 25 days** (corresponding to 10 real months in Florida). After the Florida test, the samples were characterized using thermo-mechanical analysis (DMTA; see Figure 13, Appendix A) in order to assess the effects of aging. Generally, **only very small changes in thermomechanical properties** were observed even after the longer Florida test (which simulated 10 months of aging), but specific **chemical effects of hydrochar** nevertheless could be observed. In the case of the **CB-rich** VCB40, **moderate additional crosslinking** was detected, which was more extensive and followed a different kinetic course than in VCB50 filled exclusively by CB. In the case of the **HC-rich** VCB10, a **nearly perfect stabilization of the properties** was observed.

In Figure 13a,b (*G*′ = *f*(*T*) and tan *δ* = *f*(*T*) curves), the small **changes in thermo-mechanical properties of** VCB40 are presented as examples for all **the studied materials**. The change of *G*′ (25 °C) in dependence on the aging time is analyzed in Figure 13c for all the tested samples.

**VCB40:** In Figure 13a (*G*′ = *f*(*T*) curves), it can be observed that the sample VCB40, which displayed the greatest final changes in mechanical properties during the Florida Test, undergoes slight additional crosslinking under the test conditions (see also trendline of *G*′ (25 °C) in Figure 13c). After 3.6 weeks (≡9.83 real months), the modulus *G*′ (25 °C) rises by 24%. In the region of thermal and oxidative degradation in DMTA (at T > 150 °C), the modulus of the 3.6-weeks-aged VCB40 is still slightly higher than the modulus of the intact VCB40 (Figure 13a). The shorter 1-week-Florida-Test (equivalent of 2.75 months at real conditions) also led to some additional crosslinking in VCB40: *G*′ (25 °C) increased by 11% (in contrast to VCB50, where *G*′ (25 °C) decreases after a 1 week of Florida Test). It can also be noted that between 1 and 3.6 weeks of aging (≡between 2.75 and 9.83 real months), the additional crosslinking in VCB40 was slightly slower than in VCB50.

The **hydrochar-rich sample VCB10** displays the smallest changes in DMTA as a result of Florida Test aging (see Appendix A: *G*′ *= f(T),* and tan *δ = f(T)*, and trendlines comparison in Figure 13c). The modulus *G*′ (25 °C) decreases by 5.7% after 1 week (≡2.75 real months), and then, after 3.6 weeks (≡9.83 real months), it increases again by 4.5%, so that the net modulus change relative to the intact sample is only −1.2% (a negligible decrease) after 3.6 weeks (≡9.83 real months). More visible is the modest decrease of *G′*(VCB10) in the high-temperature region above 110 °C (see DMTA in Appendix A). The value of tan *δ* (25 °C) (see Appendix A) remains unchanged after 1 week (≡2.75 real months) and drops by 13.4% of its initial value after 3.6 weeks (≡9.83 real months), which means a slight increase in elastic character.

The **hydrochar-free rubber composite VCB50** displays a smaller net effect of the aging but comparably large incremental changes in properties during the course of the Florida Test, as VCB40 does. At first, after 1 week (≡2.75 real months), the modulus *G*′ (25 °C) of VCB50 decreases by 6.3%. Subsequently, after 3.6 weeks (≡9.83 real months), it increases again by 12.4%, so a net modulus increase relative to the intact sample is achieved, specifically by 6.1% after 3.6 weeks. Above 110 °C, the modulus *G*′ is altered only very slightly by the aging (see DMTA in Appendix A). The value of tan *δ* displays only minimal changes due to the aging of VCB50: it drops by 2.6% of the original tan *δ* value after 3.6 weeks (≡9.83 real months).

## 4. Conclusions

The suitability of a new bio-sourced hydrochar (HC) as filler or co-filler in natural rubber and similar elastomers was assessed. The HC was synthesized from oak tree sawdust using a procedure previously optimized by the authors.For the above purpose, a standard vulcanized rubber recipe was modified by partially (and even fully) substituting the classical carbon black (CB) filler with HC, while the loading of the co-fillers phase was kept constant at 50 phr (31.5 wt.%).Basic mechanical and thermo-mechanical properties were analyzed via tensile and hardness tests, as well as by DMTA, swelling, and mechanical tests in the swollen state, in order to assess the effect of HC on permanent and reversible crosslinking.Due to its larger particle size and, hence, smaller specific surface area, large amounts of HC were found to reduce the permanent crosslinking density in the composites. Similar to CB, HC was found to form a considerable amount of reversible (physical) crosslinks with the elastic chains of the rubber matrix, in addition to covalent crosslinks.The chemical effects of HC were studied using long-time vulcanization experiments, TGA, as well as by thermo-oxidative aging tests at a constant, elevated temperature in circulating air (180 °C).The chemical effects of HC were interesting; if it was used as the exclusive filler component, it displayed a very strong anti-oxidizing effect, which greatly stabilized the rubber composite against oxidative crosslinking (and embrittlement).Composites with HC/CB ratios 20/30 and 10/40 displayed interesting synergic chemical stabilization and fairly good mechanical properties.HC was also found to affect the vulcanization kinetics in different ways, depending on the HC/CB ratio.Simulated weathering (industrial ‘Florida test’) led only to modest dynamic-mechanical (DMTA) properties changes. At larger HC content, oxidative crosslinking suppression became visible in the Florida test.To conclude, the obtained results indicate that HC could be a promising filler material in organic polymers due to its specific chemical reactivity (anti-oxidizing properties, radical reactions, among others).

## Data Availability

Not applicable.

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
