# Peer review of "Natural Rubber Composites Using Hydrothermally Carbonized Hardwood Waste Biomass as a Partial Reinforcing Filler—Part II: Mechanical, Thermal and Ageing (Chemical) Properties"

_polymers, 2023, doi:10.3390/polym15102397_

Round 1

Reviewer 1 Report

The article presented by the authors is interesting and constitutes a very important development of the research topic concerning elastomeric biocomposites. The overall quality of the article is high, with the scientific novelty well presented. Noteworthy is the fact that this is a 2-part article, with a lot of research methods and results.

My most important comment concerns the presentation of test results for composites filled with pure hydrochar, without the addition of soot, which I have already mentioned during the review of the first part of the article. The authors should add the obtained results for this type of system. Undoubtedly, it will complement the whole research and, above all, it will show the potential of this type of system in elastomeric composites.

The second comment concerns the clarification of the conditions that were used in the case of thermo-oxidative aging tests - paragraph 2.9. What are the reasons why the authors applied such conditions? Also, please change the nomenclature used - from "oxidation test" to "thermo-oxidative aging test".

The third remark is a request to add errors in the case of mechanical properties tests - table 3.

Author Response

Reviewer #1:

The article presented by the authors is interesting and constitutes a very important development of the research topic concerning elastomeric biocomposites. The overall quality of the article is high, with the scientific novelty well presented. Noteworthy is the fact that this is a 2-part article, with a lot of research methods and results.

ResponseThe authors are very grateful for the positive assessment, for the valuable suggestions which helped to improve the Manuscript and make it more attractive, and also for revising both parts of the submitted work.

The changes and added texts in the revised Manuscript are highlighted with yellow color in the uploaded “PDF version with highlights”. Captions of newly added Figures are highlighted in red. Captions of Figures with added new data are partly highlighted in red.

Reviewer #1:

(Point 1) My most important comment concerns the presentation of test results for composites filled with pure hydrochar, without the addition of soot, which I have already mentioned during the review of the first part of the article. The authors should add the obtained results for this type of system. Undoubtedly, it will complement the whole research and, above all, it will show the potential of this type of system in elastomeric composites.

Response 1This is a sound and valuable suggestion, for which the authors are very grateful (and for Reviewer #1 insisting on this point).

In order to study the effect of hydrochar (and of HC/CB combinations) more profoundly, the authors prepared two additional samples: one filled solely with hydrochar (with 50 phr, code “VCB00”) and one without any fillers, but with all the other additives from the rubber recipe (name: “matrix”, used as reference sample with no filler–matrix interactions). All the characterizations, except the difficultly accessible Florida Test (in which the effects on material properties generally were modest) were performed for these new samples, including vulcanization rheology and morphology study by TEM.

Interesting new insight into the physical and chemical effects of the hydrochar (HC) in the rubber composites was obtained. The additional results really were worth the effort.

Reviewer #1:

(Point 2) The second comment concerns the clarification of the conditions that were used in the case of thermo-oxidative aging tests - paragraph 2.9. What are the reasons why the authors applied such conditions? Also, please change the nomenclature used - from "oxidation test" to "thermo-oxidative aging test".

Response 2:  In the revised Manuscript, the reason for using the ‘harsh oxidizing conditions’ are now better explained. The choice of the experiment setup, as well as of the parameters, was based on the authors’ previous experience with the study of oxidation resistance of polyoxypropylene-based polymer networks. The test setup led to relatively quickly obtained results, and to big differences between the tested polymer structures. A literature citation about the use of similar tests was added (citation [31] in the revised Manuscript: Kashi, S.; De Souza, M.; Al-Assafi, S.; & Varley, R. Understanding the effects of in-service temperature and functional fluid on the ageing of silicone rubber. Polymers 11(3) (2019) 388), as well the citation of the authors work, where the method was developed (citation [32] in the revised Manuscript: Rodzeń, K.; Strachota, A.; Ribot, F.; Matějka, L.; Kovářová, J.; Trchová, M.; & Šlouf, M. Reactivity of the tin homolog of POSS, butylstannoxane dodecamer, in oxygen-induced crosslinking reactions with an organic polymer matrix: Study of long-time behavior. Polymer Degradation and Stability 118 (2015) 147).

 As suggested by Reviewer 1, we also replaced “Oxidation test” with “Thermo-oxidative aging test”.

Reviewer #1:

(Point 3) The third remark is a request to add errors in the case of mechanical properties tests - table 3.

Response 3 The values of standard deviation were added for the results of the mechanical tests in Table 3, as suggested.

Reviewer 2 Report

This paper presents the effect of co-fillers ‘hydrochar’ (HC), obtained by hydrothermal carbonization of hardwood sawdust, and commercial carbon black (CB) on the properties of natural rubber composites. The obtained composites were studied by hardness-, tensile-, DMTA and TGA analyses, as well as swelling tests.

The idea of this work might be interesting; however, the results and discussion present in the manuscript are not meaningful. The topic of the paper is directed to a narrow audience. The results obtained are not impressive, they are simply predictable. Moreover, the introduction is written with poor information and did not cover the importance of this topic. So, it should be extended with further recent literature. This is reflected in only 20 literature positions in reference section. The work should also include more information about the experiments (e.g. thickness of the vulcanizate plates or crosshead speed value for tensile tests). The English level should be improved as well.

Generally, in my opinion, instead of dividing the research into 2 articles, one could have been created with a greater substantive value, combining the research results with a clearly formulated research hypothesis (which is missing here). In the current work, a lot of information seems out of context and requires jumping into another work of the authors, which is not very pleasant for the reader.

Considering above-mentioned issues, the Reviewer believe that present paper does not fill the standards for high quality scientific articles and stand with his original decision about rejection of this manuscript for publication in Polymers journal.

Author Response

Reviewer #2:

This paper presents the effect of co-fillers ‘hydrochar’ (HC), obtained by hydrothermal carbonization of hardwood sawdust, and commercial carbon black (CB) on the properties of natural rubber composites. The obtained composites were studied by hardness-, tensile-, DMTA and TGA analyses, as well as swelling tests. The idea of this work might be interesting;

ResponseThe authors thank for the critical review of the original manuscript, for highlighting its weak points and helping to improve it, and make it much more attractive for the reader.

The changes and added texts in the revised Manuscript are highlighted with yellow color in the uploaded “PDF version with highlights”. Captions of newly added Figures are highlighted in red. Captions of Figures with added new data are partly highlighted in red.

Reviewer #2:

however, the results and discussion present in the manuscript are not meaningful.

ResponseThe authors politely disagree. Although the original manuscripts had some shortcomings, the results do not appear “not meaningful” to the authors. The studied hydrochar (HC) filler displays interesting chemical effects, like anti-oxidizing and radical reactivity. It might be hence an interesting additive for organic polymers. Also, in spite of the relatively large grains (if compared with carbon black, CB), HC was shown to possess a surprisingly large surface area available for permanent and reversible crosslinking in polymer composites. These aspects are now much better presented in the revised Manuscript.

Reviewer #2:

The topic of the paper is directed to a narrow audience.

ResponseThe authors politely disagree. The work might be of interest also for a broader audience, as it is dedicated to a comprehensive characterization of the physical and chemical interactions between polymer matrix and a new micro/nano-filler. The employed methodology might be inspiring for elucidating the physico-chemical and chemical properties of new composites/nanocomposites in general.

Reviewer #2:

The results obtained are not impressive, they are simply predictable.

ResponseThe authors politely disagree. As already mentioned further above, the studied hydrocghar (HC) displayed interesting and potentially useful chemical effects (especially the anti-oxidizing ones) and it also was shown to possess a surprisingly high surface area available for permanent and reversible crosslinking in polymer composites, in spite of relatively large grains. These properties were not “simply predictable”. In view of its properties, HC might be an interesting filler for organic polymers, with some development potential.

These aspects are now much better presented in the revised Manuscript.

Reviewer #2:

Moreover, the introduction is written with poor information and did not cover the importance of this topic. So, it should be extended with further recent literature. This is reflected in only 20 literature positions in reference section.

ResponseThe Introduction indeed was somewhat modest. In the revised Manuscript, the Introduction part was improved and expanded, including 11 additional citations. The added and modified texts are highlighted in yellow.

Reviewer #2:

The work should also include more information about the experiments (e.g. thickness of the vulcanizate plates or crosshead speed value for tensile tests).

ResponseThe missing information was added (vulcanizate thickness = 3.0 mm, crosshead speed in tensile tests = 500 mm/min). The whole Experimental Part was checked, and some further improvements were done.

Reviewer #2:

The English level should be improved as well.

ResponseWhile the authors have considerable experience in writing scientific papers in English, there admittedly were some text sections in the original Manuscript, which were not reader-friendly. The whole text was thoroughly revised and improved, in order to systematically improve the reader friendliness.

Reviewer #2:

Generally, in my opinion, instead of dividing the research into 2 articles, one could have been created with a greater substantive value, combining the research results with a clearly formulated research hypothesis (which is missing here).

Response 2Initially, the authors’ intention was to publish the research as a single paper. However, due to the large number of research methods and results, the paper became very lengthy, prompting the decision to split it into two articles: the first article (already published) has 18 pages of Manuscript + 18 pages of Supplementary Information, while the second, the currently revised one, has 32 pages of Manuscript + 8 pages of Supplementary Information. Making a merged paper hence would require to put a vast part of Results and Discussion into the Supplementary File.

The presented research is not theoretical, or aiming at verifying a theory, so it does not have a “research hypothesis” in the proper sense. It is a work dedicated to synthesis and to exploration of properties. In this context, the aim was

to assess the suitability of the bio-sourced hydrochar (HC), obtained using a procedure previously optimized by the authors, as filler (or co-filler) in natural rubber and similar elastomers,

for which purpose comprehensive physico-chemical and chemico-physical (oxidation, ageing) characterization was done. This context is now much better explained in the Abstract, Introduction / Aims of work, as well as in the Conclusions.

Reviewer #2:

In the current work, a lot of information seems out of context and requires jumping into another work of the authors, which is not very pleasant for the reader.

Response 2This aspect indeed was a considerable weak point of the original Manuscript. The authors thoroughly revised the Manuscript in order to make it really ‘free standing’, by adding new discussion, but also new results (including new samples). When reading the revised Manuscript, there is now no need to jump into the previous work.

Reviewer #2:

Considering above-mentioned issues, the Reviewer believe that present paper does not fill the standards for high quality scientific articles and stand with his original decision about rejection of this manuscript for publication in Polymers journal.

Response 3The authors think that the revised Manuscript now fulfills the standards for high quality scientific articles, and that it presents results which are of interest for the readers of Polymers.

Reviewer 3 Report

            Strachota et al investigate composites based on waste biomass and natural rubber. Then, the mechanical, thermal, and aging properties were investigated. The main motivation of the work was to partially replace traditional fillers such as CB with hydrochar. The final results show that the rubber composites with 10-20 phr of CB replaced with HC might be promising for overall properties improvements. However, the paper can be reconsidered following revisions. Some points are –

[1] The introduction can be improved further by discussing more literature that deals with the subject of interest of the present paper. Such as [https://doi.org/10.1002/app.44407], [https://doi.org/10.5254/rct.13.87903].

[2] In section 2.1, please provide the chemical purity of the materials used in this work.

[3] In section 2.4 -vulcanization, how authors optimize the curing condition. This should indeed be supported by rheometric curves. Please check.

[4] Please provide a study on the morphology of fillers by SEM. Please also study the filler dispersion and correlate them with the properties of composites.

[5] All of the figures need to be improved. Please add data from the control sample or unfilled NR, composite with 50 phr of HC. Then, made a comprehensive comparative study of these composites.

Good Luck with the revisions!

Author Response

Reviewer #3:

Strachota et al investigate composites based on waste biomass and natural rubber. Then, the mechanical, thermal, and aging properties were investigated. The main motivation of the work was to partially replace traditional fillers such as CB with hydrochar. The final results show that the rubber composites with 10-20 phr of CB replaced with HC might be promising for overall properties improvements. However, the paper can be reconsidered following revisions. Some points are –

Response The authors are very grateful for the generally positive assessment, and for the valuable suggestions which helped to improve the Manuscript and make it more attractive.

The changes and added texts in the revised Manuscript are highlighted with yellow color in the uploaded “PDF version with highlights”. Captions of newly added Figures are highlighted in red. Captions of Figures with added new data are partly highlighted in red.

Reviewer #3:

(Point 1) The introduction can be improved further by discussing more literature that deals with the subject of interest of the present paper. Such as [https://doi.org/10.1002/app.44407], [https://doi.org/10.5254/rct.13.87903].

Response 1As suggested by Reviewer #3, the Introduction part was improved and expanded. The added and modified texts were highlighted in yellow. Besides the citations mentioned by the Reviewer (now [7a] and [7b] in the revised Manuscript, dealing with filler geometry and with filler-matrix interactions, respectively), 11 new citations also were added.

Reviewer #3:

(Point 2) In section 2.1, please provide the chemical purity of the materials used in this work.

Response 2 As suggested by Reviewer #3, the chemical purity of the used materials is now mentioned in the revised Manuscript (lines 135–140 on p.3/4) :

“As presented in Table 1, the additives used in this work are: N-isopropyl-N’-phenyl-p-phenylenediamine (IPPD; anti-oxidant and anti-ozonant), degree of purity 97%; zinc oxide (ZnO; simple vulcanization accelerator), degree of purity 99.9%; stearic acid (dispersing agent for ZnO), degree of purity 97%; sulfur (crosslinker), degree of purity 99.999%; and N-cyclohexylbenzothiazol-2-sulfenamide (CBS; vulcanization accelerator with delayed action), degree of purity 95%.”

Reviewer #3:

(Point 3) In section 2.4 -vulcanization, how authors optimize the curing condition. This should indeed be supported by rheometric curves. Please check.

Response 3 This is indeed an interesting aspect. The authors added a discussion of the cure rheology to the first section of the Results and Discussion part (“3.1. Rubber recipe mixing, Morphology and Vulcanization”, including Figure 2). The cure behavior of all the tested compositions is now discussed, as well as the selection of the standard cure procedure for all the samples (see p. 8–10, lines 331-406).

Reviewer #3:

(Point 4) Please provide a study on the morphology of fillers by SEM. Please also study the filler dispersion and correlate them with the properties of composites.
Response 4:   This is indeed a valuable suggestion. The authors added a discussion of the morphology of the studied samples (see p. 7–8, lines 279–329, including new Figure 1), where the dispersion of the fillers is compared. Also the characteristics of the used fillers are mentioned and commented in this added discussion part.

Reviewer #3:

(Point 5) All of the figures need to be improved. Please add data from the control sample or unfilled NR, composite with 50 phr of HC. Then, made a comprehensive comparative study of these composites.

Response 5:  This is a sound and valuable suggestion, for which the authors are very grateful. A similar suggestion was made also by Reviewer #1.

As suggested, in order to study the effect of hydrochar (and of HC/CB combinations) more profoundly, the authors prepared two additional samples: one filled solely with hydrochar (with 50 phr, code “VCB00”) and one without any fillers, but with all the other additives from the rubber recipe (name: “matrix”, used as important reference sample with no filler–matrix interactions). All the characterizations, except the difficultly accessible Florida Test (in which the effects on material properties generally were modest) were performed for these new samples (including vulcanization rheology and morphology study by TEM) and the results were added to the Figures.

Interesting new insight into the physical and chemical effects of the hydrochar (HC) in the rubber composites was obtained. The additional results really were worth the effort.

Reviewer #3:

Good Luck with the revisions!

ResponseThe authors thank for the friendly closing remark, and also for the time spent with revising the Manuscript, as well as for the very useful ideas about improving the paper.

Round 2

Reviewer 1 Report

The authors made diligent revisions to the manuscript in accordance with the comments. In my opinion, the article is ready for publication in its present form.

Reviewer 2 Report

The reviewer accepts significant corrections provided by the authors and their arguments for the concerns about this manuscript.

Reviewer 3 Report

Accept in present form